# Residual Magnetic Field Non-Destructive Testing of Gantry Cranes

**DOI:** 10.3390/ma12040564

**Published:** 2019-02-14

**Authors:** Janusz Juraszek

**Affiliations:** Faculty of Materials, Civil and Environmental Engineering, University of Bielsko-Biala; 43-309 Bielsko-Biala, Poland; jjuraszek@ath.bielsko.pl; Tel.: +48-33-8279191

**Keywords:** gantry crane, RMF technique, civil engineering

## Abstract

Non-destructive tests of gantry cranes by means of the residual magnetic field (RMF) method were carried out for a duration of 7 years. Distributions of the residual magnetic field tangential and the normal components of their gradients were determined. A database of magnetograms was created. The results show that the gradients of tangential components can be used to identify and localize stress concentration zones in gantry crane beams. Special attention was given to the unsymmetrical distribution of the tangential component gradient on the surface of the crane beam No. 5 (which was the most loaded one). The anomaly was the effect of a slight torsional deflection of the beam as it was loaded. Numerical simulations with the finite element method (FEM) were used to explain this phenomenon. The displacement boundary conditions introduced into the simulations were established experimentally. Validation was carried out using the X-ray diffraction method, which confirmed the location of strain concentration zones (SCZs) identified by means of RMF testing.

## 1. Introduction

The analysis of classical standard-related crane tests makes it possible to offer a general assessment of the gantry crane technical condition. This is usually a *post-factum* activity, i.e., an activity performed after the occurrence of a crack or damage. In the case of overhead cranes operated for over 30 years, a different, more thorough assessment is necessary. Standard tests are unable to determine whether stress concentration zones that cause cracks have appeared in the crane structure. This paper presents a new approach that enables a priori diagnostics, allowing the identification of potentially hazardous areas in advance.

The residual magnetic field (RMF) has been used as an inspection tool in transportation, power engineering and in the metal industry [1]. For a ferromagnetic structure, such as a crane, the RMF technique is expected to be one of the solutions that may enable early damage evaluation. It is proved in [2] that the size of the sample does not change the magnetization curve profile but it affects the RMF value. Four sets of Q345 steel samples with different widths and thicknesses were tested in the laboratory. The experimental results are explained by the theory of the interaction between dislocation and the domain wall, as well as the theory of the demagnetizing field. It is proved in [3] that the size of the sample does not change the magnetization curve profile but it affects the RMF value. Four sets of Q345 steel samples with different widths and thicknesses were tested in the laboratory. The experimental results are explained by the theory of the interaction between dislocation and the domain wall, as well as the theory of the demagnetizing field.

Significant static and fatigue experiments were performed with the use of the RMF technique [4,5,6,7,8,9,10,11,12,13,14]. There are numerous publications on the analysis of damage identified in flat specimens using the RMF technique. For example, damage identification in flat 12-mm-thick specimens subjected to tensile stresses was analyzed in [15] by Shui et al. The places in which the normal component Hn is zero indicate locations of damage to the specimen. The RMF also enables the identification of stress concentration zones. An interesting work on this subject is [16], which discussed the impact of the occurrence of stress concentration zones in three-point and four-point bending tests on the distribution of the Hp and Hn components, as well as the reduced stress according to the Huber–Misses–Hencky hypothesis. The RMF technique is used in the diagnostics of ferromagnetic structural elements. The RMF enables the detection of cracks, micro-cracks and closed cracks, which are difficult to detect using traditional methods. Knowing the previous magnetic image of the examined component, it is possible to detect stress concentration zones before actual cracks or micro-cracks appear. It is one of few methods that indicates, a priori, hazardous areas in the structure [17].

## 2. Materials and Methods

The analyzed gantry crane works in an open area and is used to transport coal from a storage site to a furnace. It was built in 1980. The structure consists of a truss bridge rolling on rails installed on two parallel 13-span assemblies consisting of crane beams. One span is less than 12 m long. The width of the crane is 32.01 m, and the trestle bridge reaches a height of 8.32 m (Figure 1). The cross-section of the crane beam is designed as an I-bar with a height of 824 mm. The web was made of sheet metal with a width of 800 mm and a thickness of 10 mm, while the feet were made of sheet metal with a width of 250 mm and a sheet thickness of 12 mm. 

The approximate weight of the crane carried by the beam and the maximum load weight that can be transported by the structure were taken into account. The load was divided into two spot forces acting in the place where the gantry crane wheel rests on the beam. The total operating load was Fo = 40 kN, while the crane structure weight load was 150 kN distributed over two road wheels, 75 kN each. At this load, the highest stress total was 40.5 MPa.

### RMF Methods

In general, local irregularities in the material homogeneity resulting from stress, plastic strain or fatigue-induced changes in the structure generate corresponding, as well as local, changes in the degree of the material magnetization, and thus also in induction. In this situation, an induction component perpendicular to the surface of element $B_{p}$ appears, which results in the creation of a vertical magnetic field component $H_{p}$ outside the element, the distributions of which are recorded by the probes used in the RMF method. According to Equation (1):(1)$$H_{p}=\frac{1}{\mu_{o}}B_{p}$$
where:$B_{p}$ is the induction component perpendicular to the metal surface;$\mu_{0}=2\pi *10^{-7}$ is the air magnetic permeability.

Since $B_{p}$ and thus $H_{p}$ are proportional to the rate of changes in induction *B* along the tested element, it can be assumed that:(2)$$H_{p}\sim \frac{1}{\mu_{o}}\frac{\Delta B}{\Delta z}$$
where: Δz is the length at which induction B changes.

Juraszek [7] claims that when entering the material area with a permeability μ lower than in the rest of the material, e.g., in the area of a strong plastic strain, the induction flux is scattered and the RMF vertical component above the surface has a positive sign. Moving to an area of higher magnetic permeability (e.g., in the area of increased elastic stresses), the lines of the magnetic field inside the material are focused and the vertical component $H_{p}$ has a negative sign. Away from areas of different magnetic properties, when the material is homogeneous, the residual magnetic field disappears. In this method, the RMF component $H_{p}$ is measured by determining the following components: tangential component $H_{p\left(x\right)}$; normal component $H_{p\left(y\right)}$.

Based on [11,14], Equation (3) and Equation (4) respectively present simplified formulae describing the values of components $H_{p\left(x\right)}$ and $H_{p\left(y\right).}$ The values have been confirmed experimentally.
(3)$$H_{x}=\int_{0}^{b}dH_{x}=\frac{1}{2\pi \mu_{0}}\int_{0}^{b}\rho \left(l\right)*\frac{\left(x-l\right)}{{\left|\left|r\right|\right|}^{2}}dl$$
(4)$$H_{y}=\int_{0}^{b}dH_{y}=\frac{1}{2\pi \mu_{0}}\int_{0}^{b}\rho \left(l\right)*\frac{\left(y\right)}{{\left|\left|r\right|\right|}^{2}}dl$$
where:
r=[x−l,y] is a vector, a variable related to the element surface;ρ is a function defining the boundary condition presenting the distribution of the magnetic charge density on the specimen surface, which depends directly on the dislocation density and indirectly on the stress;*x*, *y* are the coordinates of the point on the tested element surface;*l* is the dislocation length;*b* is the dislocation width.

In the RMF method, the so-called magnetic stress intensity factor, or the gradient of the magnetic field normal component, was adopted as a measure of the quantitative assessment of the stress concentration level. The gradient of the residual magnetic field is defined by Equation (5):(5)$$K_{in}=\frac{\left|\Delta H_{p}\right|}{2l_{k}}$$
where:
$K_{in}$ is the RMF gradient;$\left|\Delta H_{p}\right|$ is the absolute value of the difference in $H_{p}$ between two control points located at equal distances $l_{k}$ on both sides of the line $H_{p}$ = 0.

Due to the magnetoelastic effect, the defective area is very easily detectable under the influence of the load. It is characterized by a different modulus of elasticity, magnetic susceptibility or magnetoelastic properties. Therefore, under the influence of the load, its deformation value as well as the magnetoelastic increase in magnetization will be different from that of the matrix. 

The difference in magnetization between the defective area and the matrix can be described using the following formula:(6)$$\Delta M=\Delta M\left(H_{0}\right)+\Delta M^{\sigma }\left[H_{0};\sigma \left(x,\alpha \right)\right]$$
where: $\Delta M\left(H_{0}\right)$ is the rise in magnetization under the influence of a weak magnetic field;$\Delta M^{\sigma }\left[H_{0};\sigma \left(x,\alpha \right)\right]$ is the rise in magnetization under the influence of elastic stresses in the presence of a weak magnetic field.

The result is a clear increase in the residual magnetic field within the defect area, since its value is directly proportional to ΔM.

## 3. Results of the RMF Scanning

Span No. 5 from the set of 13 spans was selected for analysis. The span is located directly above the coal chute to the conveyor that transports fuel to the turbine boiler (cf. Figure 1, showing a diagram of a single span). Each coal load is moved to span No. 5 using the crane. The crane with the load must stop on span No. 5, causing additional dynamic loads. Then, coal is put into the chute, which in turn causes vibrations of the crane beam. Due to this, the load of span No. 5 is the highest in the entire crane. The measurements involved scanning the bottom surface of the crane beam and determining the distributions of the normal and tangential components of the residual magnetic field. Gradients of both RMF components were also determined. 

Next, based on the analysis of the gradient distribution, the highest gradient value of 10 A/m/mm was adopted as the limit criterion based on numerous previous works by the author [7,11]. Areas where the limit gradient value was exceeded were selected. They were then subjected to a further detailed analysis. The beam measurements were carried out in two stages.

The first stage of the analysis consisted in scanning the entire underside of span No. 5 along the previously determined three uniformly distributed measuring lines L1, L2, L3, as shown in Figure 2. It should be noted that line 3 is placed on the chute side.

Example results of the beam surface scanning in the form of distributions of the tangential and the normal component (Hpx and Hpy, respectively) are shown in Figure 3 and Figure 4 below. The value of the tangential component along the length of measuring line 3 ranges from −100 A/m to 69 A/m. The value of the orthogonal component is in the range of 340 A/m to −80 A/m. The gradient of the tangential and normal components is many times higher than the limit value of 10 A/m/mm.

This section provides a concise and precise description of the experimental results, their interpretation as well as the conclusions that can be drawn from the experiments. In the second stage, the area with the highest gradient value was selected. For this purpose, the measurements were first carried out with no coal load on the crane, and then the operation was repeated after the crane bucket was filled with coal. In both cases, based on the analysis of the obtained magnetograms and comparing the gradient values of the RMF component Hp with the limit value of 10 A/m/mm, the measurements demonstrated the occurrence of stress concentration zones between 5655 and 7145 mm from the crane left support. Then, the most loaded part of the beam, indicated in red in Figure 5, was tested again. An additional 22 measuring lines with numbers from L27 to L49 were introduced in this area.

The distribution of the changes in the residual magnetic field tangential component gradient for measuring line 3, for the selected sub-area, is presented in Figure 6.

The adopted procedure was a gradual refinement of the observation area. Based on the analysis of the tangential component gradient distribution, stress concentration zones were found to occur in places where the limit of 10 A/m/mm was exceeded, as indicated in earlier studies by the author [7]. These are areas covering measuring lines 36–38, while the second area covered lines 40–45. In the next stage of the analysis, 2-D maps of the distribution of the magnetic field component gradients were prepared. The data for the residual magnetic field intensity maps were imported from J&J System software, while the gradients were calculated as a derivative of the RMF component $H_{p}$ over the measurement line length using the finite difference method according to Equation (7):(7)$$G=H_{p}\left(x\right){}^{\prime }\approx \frac{H_{p}\left(x+a\right)-Hp\left(x\right)}{a}$$
where: Hp(x) is the RMF intensity value in the point where the gradient was determined;$H_{p}\left(x+a\right)$ is the RFM intensity value for the next measuring point;*a* is the distance between consecutive points on the measuring line. 

It was decided that for the magnetic field intensities, the colors would change every 10 units. The intensities of 0–10 A/m are presented as white. The values of positive intensities depending on the value are represented by warm colors from yellow to brown, while negative values are represented by cold colors from green to purple. For the gradient maps, colors changed every 1 unit. Because the gradient is non-negative, the colors range from white to dark brown. The distribution of the RFM tangential component on the 2-D map for the selected area is shown in Figure 7. It can be noticed that the highest values of 130 A/m were in the range of 607 to 1290 mm (*x*) and 152 to 380 mm (*y*). A relatively small area was found for the coordinates of 480 (*x*) and 16 mm (*y*). 

Further analyses in the selected area concern the distribution of the gradient of the tangential and normal components. Figure 8 presents the distribution of the tangential component gradient for the full load on the crane beam.

If the crane is loaded with a truck only, the gradient values are lower by approximately 2 A/m/mm. Areas with values exceeding 10 A/m/mm also coincide with the areas of the highest values of the tangential component. They are located in the upper left quadrant of the map in the range of 607–1290 mm (*x*) and 140–380 mm (*y*). This area is located on the coal chute side—the analyzed part of the beam is located exactly above the coal chute. It is also the most heavily loaded beam, because during each cycle coal is transported at various points of the coal storage, always above the chute, where the load of a full skip hoist is applied first, and then rapid unloading takes place by opening the skip hoist and pouring out the coal. The estimated number of cycles in the heating season is about 60,000. A similar distribution of the normal component gradient confirmed the occurrence of the identified stress concentration zones. The highest values of the gradient of the RMF normal component reached 14 A/m/mm. Figure 9 presents the normal component gradient distribution. The area with the highest gradient value appears in the top left quarter of the 2-D map. 

## 4. Validation and Numerical Simulation

During the crane beam overhaul, samples were taken to determine the distribution of stress components. The samples were collected from places indicated as SCZs. The tests were carried out by means of the X-ray angle diffraction using an AvanceD-8 diffraction instrument. The testing results confirmed stress concentration zones determined using the RMF method. Stresses in the SCZs reached a value of 186 MPa, and outside the zone, of about 60 MPa. The diffraction pattern is shown in Figure 10. 

The detailed and comprehensive experimental and numerical verification consisted in the determination of deflection values halfway through the span length. The values were determined in three locations (points) marked in the figure presenting the crane beam cross-section (cf. Figure 11).

Figure 12 presents influence lines for span No. 5 determined experimentally (values measured during the gantry crane passage). The measuring points located on the side of the coal chute (inner points) were characterized by the biggest value of deflection of 7.9 mm, whereas the smallest deflection value of 5.2 mm was observed for the opposite (outer) points. The deflection value of the measuring point above the I-beam web (the central point) was included between the values found for the outermost points. It totaled 6.35 mm. The deflection values determined in the crane beam measuring points gave evidence of beam warp. It turns out that this is the effect of the eccentric load of the analyzed crane beam I-section (which does not comply with the gantry crane technical documentation). 

The experimentally determined values were entered into the FEM numerical model. The element side characteristic length was 2–10 mm. Shell 63 element was used. As the aim of the analysis was to determine the deformation of the crane beam and establish the cause of the unsymmetrical distribution of the tangential component gradients at set boundary conditions, a linear elastic material model was used. The constitutive model was linear. The total number of elements was 1.4 million. The numerical testing results confirm the twisting of the beam, which causes a higher level of stress. The experimental results were introduced into the FEM numerical model in the form of boundary conditions for displacement. A 3-D numerical model of the analyzed crane girder was built based on the finite element method ANSYS code, taking into account the upgrades and the span axis displacement detected in relation to the half-distance between the truss poles. The results of the numerical simulations (Figure 13) confirmed the previously indicated two areas, i.e., the area in the middle of the span length and in the places of welding the reinforcing cover. The stress levels in the analyzed areas were also determined. In the middle of the span length, the stresses totaled ~40 MPa.

The reason for this is the slight displacement of the crane travelling wheel towards the coal chute. The determined numerical distribution of the crane beam displacement is shown in Figure 11.

## 5. Conclusions


The RMF technique enabled the effective location of stress concentration zones in the analyzed crane beam by determining the gradient of the RMF tangential component.The experimental verification using the X-ray diffraction method confirmed the stress concentration zones in the beam.Creating a relational database of magnetograms of the ferromagnetic structure from the beginning of its operation seems to be an interesting addition to standard magnetic diagnostic methods.Combining different methods and measurement techniques, for example a fiber optic beam deflection measurement system using the FEM method which implements boundary conditions from optical fiber measurements, with residual magnetic field techniques can significantly contribute to the identification of stress concentration zones.


## Figures and Tables

**Figure 1 materials-12-00564-f001:**
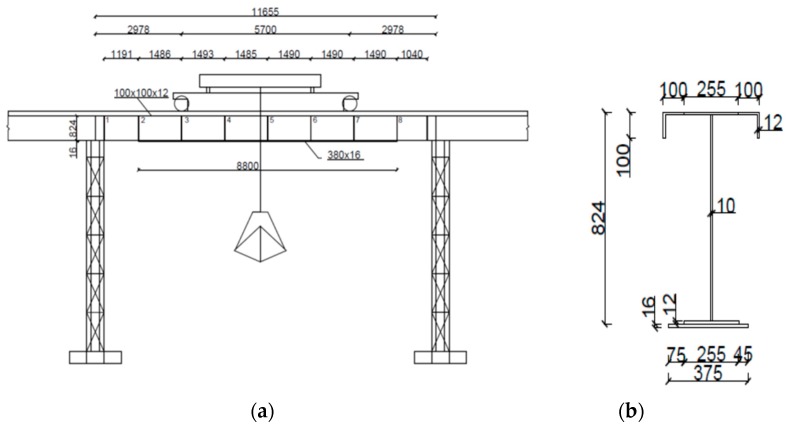
Diagram: (**a**) crane beam, (**b**) cross-section of the beam.

**Figure 2 materials-12-00564-f002:**
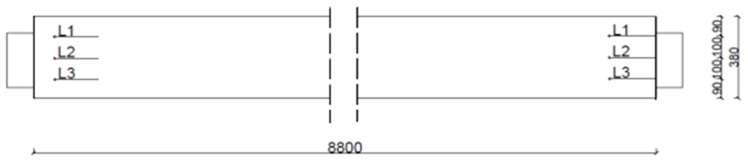
Diagram of measuring lines L1, L2, L3.

**Figure 3 materials-12-00564-f003:**
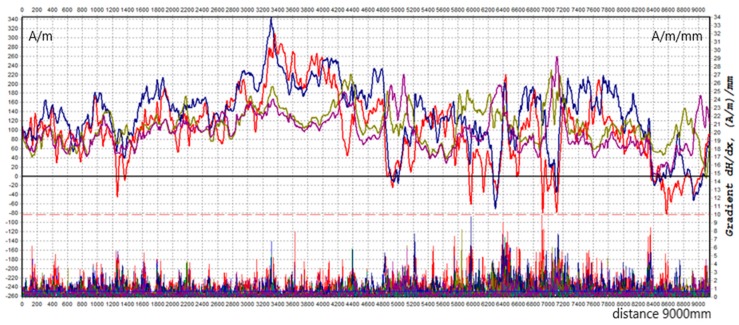
Distribution of tangential component Hpx along L3.

**Figure 4 materials-12-00564-f004:**
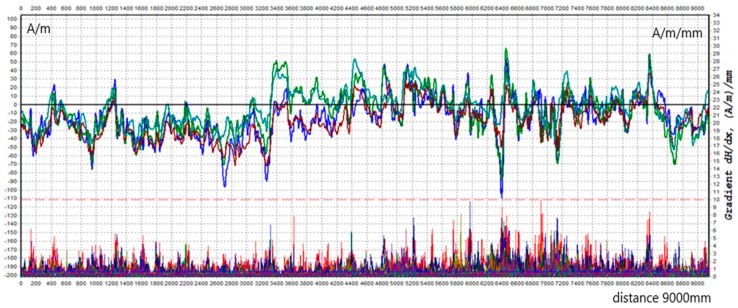
Distribution of normal component Hpy along L3.

**Figure 5 materials-12-00564-f005:**
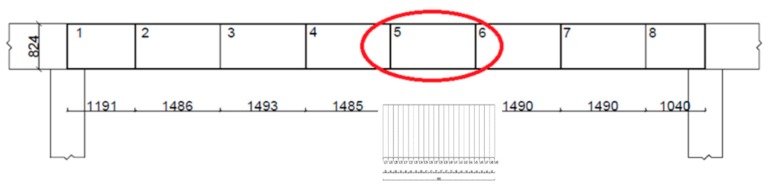
The beam’s most loaded part and the additional 22 measuring lines.

**Figure 6 materials-12-00564-f006:**
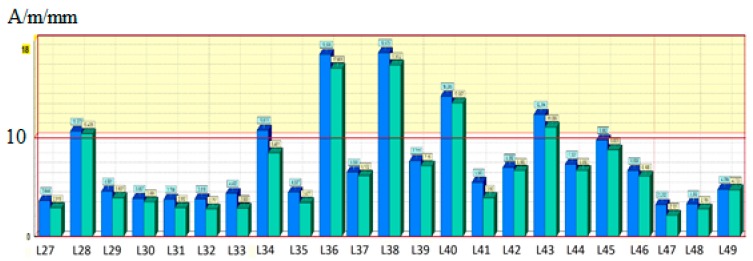
Changes in the tangential component gradient for the selected sub-area under analysis.

**Figure 7 materials-12-00564-f007:**
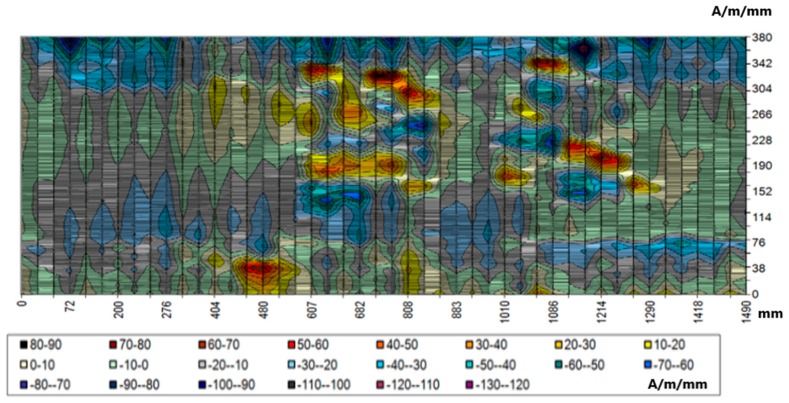
Distribution of tangential component Hpy for the selected area.

**Figure 8 materials-12-00564-f008:**
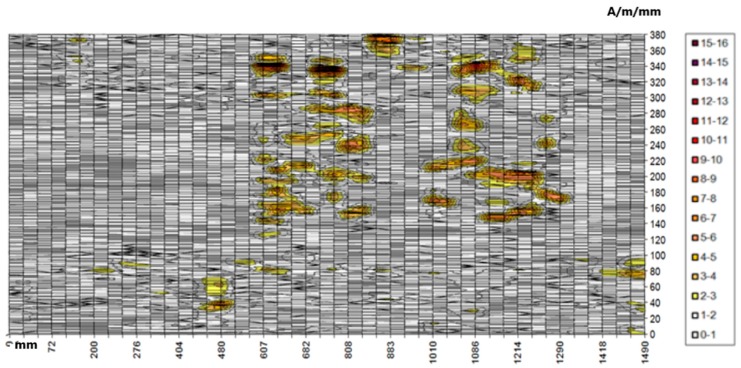
Distribution of the gradient of the tangential component dHpx/dx for the selected area.

**Figure 9 materials-12-00564-f009:**
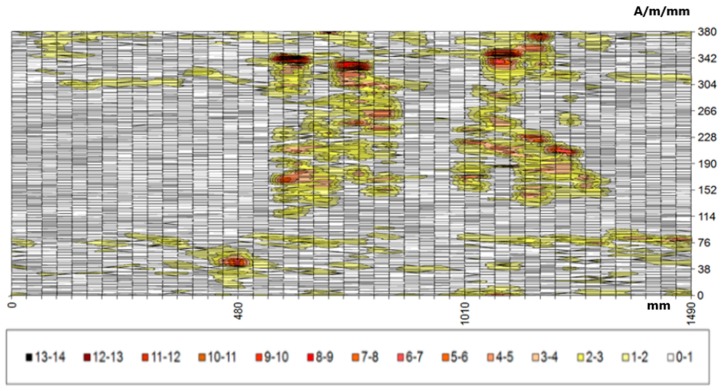
Distribution of the gradient of the normal component for the selected area.

**Figure 10 materials-12-00564-f010:**
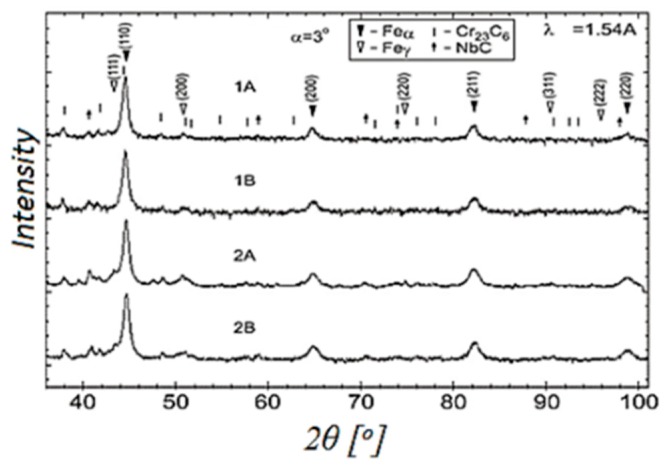
Diffraction pattern of a sample taken from the crane beam.

**Figure 11 materials-12-00564-f011:**
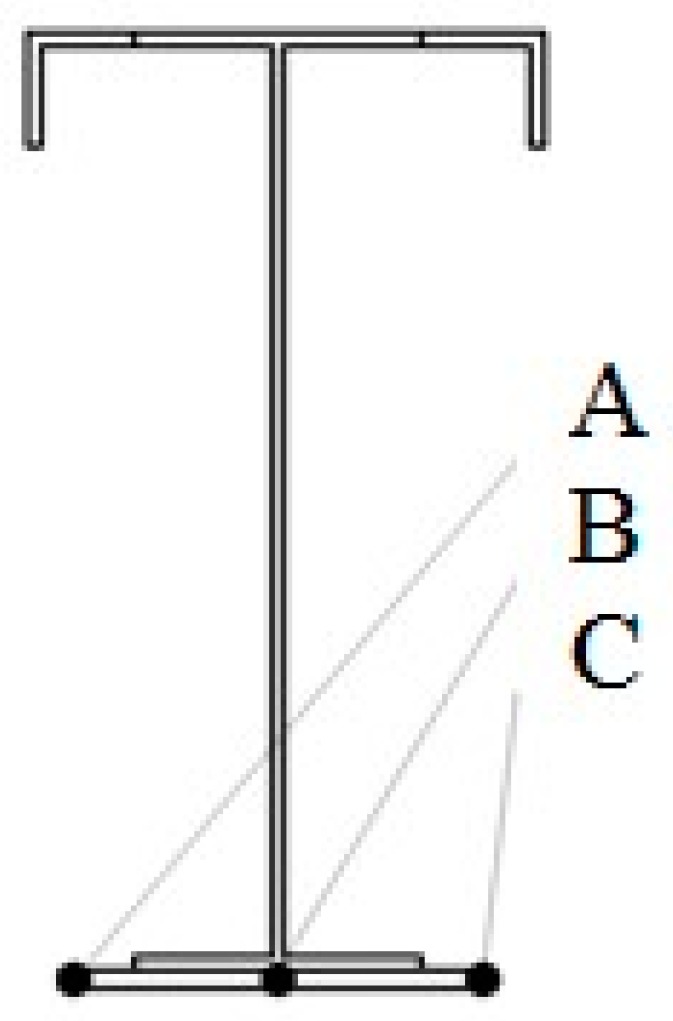
Arrangement of measuring points in the cross-section halfway through the span length.

**Figure 12 materials-12-00564-f012:**
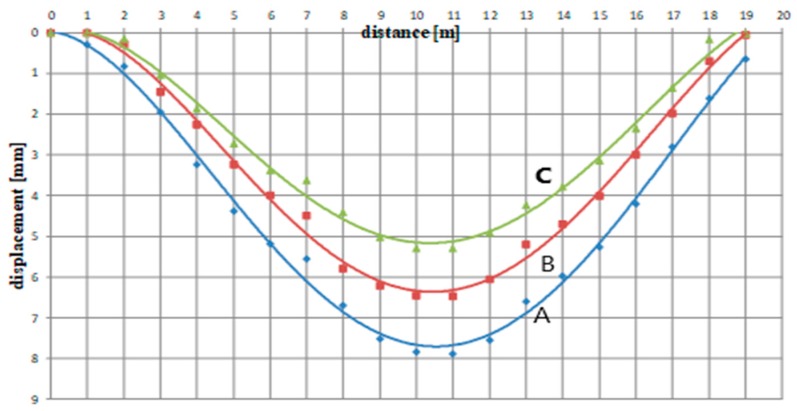
Crane beam deflection during the crane crab passage.

**Figure 13 materials-12-00564-f013:**
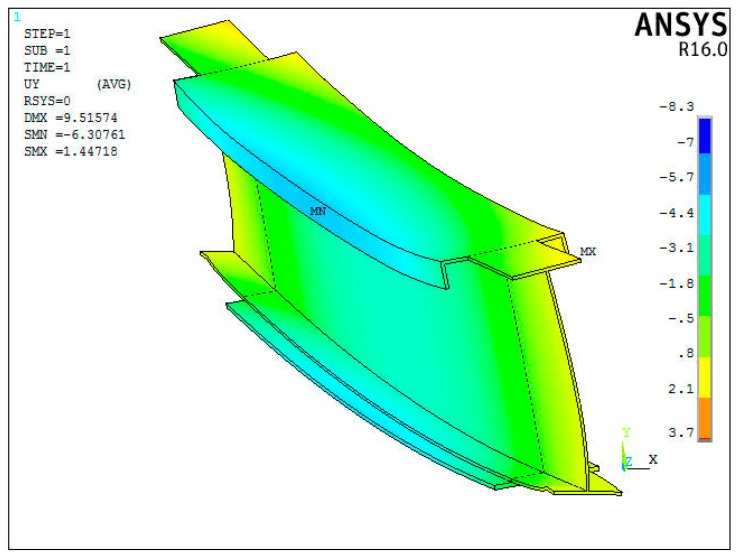
Distribution of the UY displacement.

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
