# Peer review of "Residual Magnetic Field Non-Destructive Testing of Gantry Cranes"

_materials, 2019, doi:10.3390/ma12040564_

Round 1

Reviewer 1 Report

The manuscript by Juraszek entitled “RMN Non-destructive Testing of Gantry Crane” focuses on the use of residual magnetic field technique as non-destructive procedure. The paper matches the journal aims and scope (characterization techniques) and the content is valuable and interesting. However, the following corrections and/or additional work are required:

·      English grammar, style and typos must be corrected (e.g. line 27, 218,…)

·      For the sake of clarity, the title must include not only the acronym (RMF) but also the complete name of the technique.

·      The abstract seems to be an introduction more than a summary of the main findings. Please, rewrite.

·      The pros and cons, weaknesses and strengths of the technique must be pointed out, highlighting the benefits of using it in structural characterization campaigns.

·      It is really important to indicate which is the maximum crack depth that can be detected.

·      It is necessary to review and include in the introduction section more recent advances (last five years), e.g: 

o  “Effect of sample size on the residual magnetic field of ferromagnetic steel subjected to tensile stress” 2018. Insight: Non-destructive testing and condition monitoring.

o  “Defect identification in ferromagnetic steel based on residual magnetic field measurements” 2017. Journal of Magnetisms and Magnetic Materials.

·      The citation style must be corrected, and the name of the researcher/s must be written instead of the reference number in some cases (e.g. line 39, line 79).

·      Line 64, please rewrite the sentence (it is difficult to understand it).

·      Line 133, please include the reference in the brackets.

·      Section 4 (validation and numerical simulation) must be improved: Which element type has been used? Which is the mesh density? Which is the material constitutive model? A mesh independence study has been performed? Which is the number of elements? Is the analysis linear? Which are the material control parameters? 

·      A comprehensive and detailed numerical vs. experimental validation must be provided. 

Author Response

The entire text has been revised and the Reviewer’s suggestions have been taken into account. An explanation related to the experimental and numerical verification has been added.

 New abstract

Reviewer 1

The manuscript by Juraszek entitled “RMN Non-destructive Testing of Gantry Crane” focuses on the use of residual magnetic field technique as non-destructive procedure. The paper matches the journal aims and scope (characterization techniques) and the content is valuable and interesting. However, the following corrections and/or additional work are required:

·      English grammar, style and typos must be corrected (e.g. line 27, 218,…) Done

Line 27 assessment of the gantry crane technical condition. This is usually a post-factum activity, i.e. activity performed after the occurrence of a crack or damage. In the case of overhead cranes operated for over 30 years, another, more thorough, assessment is necessary. Standard tests are unable to answer the

·      For the sake of clarity, the title must include not only the acronym (RMF) but also the complete name of the technique. Corrected

Residual Magnetic Field  Non-destructive Testing of Gantry Crane

·      The abstract seems to be an introduction more than a summary of the main findings. Please, rewrite. Done

·      The pros and cons, weaknesses and strengths of the technique must be pointed out, highlighting the benefits of using it in structural characterization campaigns. The RMF technique does not require magnetization or any special preparation of the surface. Apart from crack detection, it also makes it possible to localize stress concentration zones (SCZ’s).

·      It is really important to indicate which is the maximum crack depth that can be detected. It is possible to detect cracks with the depth of up to 1.5 mm.

·      It is necessary to review and include in the introduction section more recent advances (last five years), e.g: 

o  “Effect of sample size on the residual magnetic field of ferromagnetic steel subjected to tensile stress” 2018. Insight: Non-destructive testing and condition monitoring.

·       “Defect identification in ferromagnetic steel based on residual magnetic field measurements” 2017. Journal of Magnetisms and Magnetic Materials.

The indicated literature items have been included in the literature survey.

It is proved in [2] that the size of the sample does not change the magnetization curve profile but it affects the RMF value. Four sets of Q345 steel samples with different widths and thicknesses were tested in the laboratory. The experimental results are explained by the theory of the interaction between dislocation and the domain wall, as well as the theory of the demagnetizing field.

In [3], variations in the residual magnetic field (RMF) and in its gradients on the surface of 30CrNiMo8 steel specimens with two types of the defect shape were tested depending on tensile tests. It was found that during the loading process both the RMF and the gradient curves of the two defective specimens demonstrated similar evolution patterns in terms of shape and location. The RMF gradients exhibit more pronounced characteristics than the RMF signals in the defect area, which makes it possible to capture information on defects.

·      The citation style must be corrected, and the name of the researcher/s must be written instead of the reference number in some cases (e.g. line 39, line 79).

For example, damage identification, if only in flat, tensioned specimens with a thickness of 12mm, was analysed in [15] by Shui G.;  Li. Ch.;   Yao K.. The places in which the normal component Hn is zero indicated locations of damage to the specimen. RFM also allows the identification of stress concentration zones. An interesting work on this subject is article [16] discussing the impact of the stress concentration zones occurrence in the 3 and 4-point bending test on the distribution of Hp and Hn

And so, in [7] Juraszek claims that when entering the area of the material…

·      Line 64, please rewrite the sentence (it is difficult to understand it).

kN each. For such a static load distribution, the greatest bending stress is 40.5 MPa. At this load, the highest stresses total 40.5

·      Line 133, please include the reference in the brackets.

The span no. 5 from the set of 13 spans was selected for the analysis. It is located directly above the coal chute to the conveyor that transports it to the turbine boiler (cf. Fig. 1 – diagram of a single span)

·      Section 4 (validation and numerical simulation) must be improved: Which element type has been used? Which is the mesh density? Which is the material constitutive model? A mesh independence study has been performed? Which is the number of elements? Is the analysis linear? Which are the material control parameters? 

·      A comprehensive and detailed numerical vs. experimental validation must be provided. 

The detailed and comprehensive experimental and numerical verification consisted in determination of deflection values halfway through the span length. The values were determined in 3 locations (points) marked in the figure presenting the crane beam cross-section (cf. Fig. 6).

Fig. .    Arrangement of measuring points in the cross-section halfway through the span length

Fig. 7 presents influence lines for span No. 5 determined experimentally (values measured during the gantry crane passage). The measuring points located on the side of the coal chute (inner points) were characterized by the biggest value of deflection of 7.9 mm, whereas the smallest deflection value of 5.2 mm was observed for the opposite (outer) points.  The deflection value of the measuring point above the I-beam web (the central point) was included between the values found for the outermost points. It totalled 6.35 mm. The deflection values determined in the crane beam measuring points prove the beam warp. It turns out that this is the effect of the eccentric load of the analysed crane beam I-section (which does not comply with the gantry crane technical documentation).

Crane beam deflection during the crane crab passage

The experimental testing results were introduced into the FEM numerical model in the form of displacement boundary conditions. The numerical simulations confirmed the gantry crane beam deflection with a simultaneous twist. The beam demonstrates a complex stress state – both normal and tangential stresses occur. The latter are particularly unfavourable for such structures.

Reviewer 2 Report

-Figures should be of high quality; in case they are reproduced or reused from a citeable source, this should be referred

-Further discussion should accompany each figure, e.g. Fig. is not properly discussed

-Motivation and novelty should be added at the end of introduction

-Abbreviations in full where first used. In the beginning of sentence, they should remain in full throughout the whole text

-References should be up-to date

-How many tests were conducted?this should be reflected in given results as error

Author Response

The entire text has been revised and the Reviewer’s suggestions have been taken into account. An explanation related to the experimental and numerical verification has been added.

The text has been revised.

Figures should be of high quality; in case they are reproduced or reused from a citeable source, this should be referred

The quality of the figures will be improved as much as possible.

-Further discussion should accompany each figure, e.g. Fig. is not properly discussed The text has been changed.

-Motivation and novelty should be added at the end of introduction The motivation was implementation of the RMF method in non-destructive testing of large-size structures, such as a gantry crane, in a complex strain-and-stress state.

-Abbreviations in full where first used. In the beginning of sentence, they should remain in full throughout the whole text  Corrected

-References should be up-to date References updated

-How many tests were conducted?this should be reflected in given results as error. Thirty tests were performed. The measuring error was smaller than 3%.

Reviewer 3 Report

The main problem of the paper is that it seems more a case study than a research paper, so the level of originality, innovation and scientific interest is not high.

To my opinion, if the authors cannot effectively illustrate the innovative and research aspects of their work and the gap that the paper can fill compared to the existing literature, the paper is not suitable for publication as a research work.

In addition to this fundamental flaw, the paper presents a series of aspects that should be improved or corrected:

1) The description of the state of the art is meagre, the bibliographical references are few and limited, especially those related to the theoretical aspects of RFM applied in the section 2. There is also a lack of references to important works in the same field, which would help to understand well the topic covered and the usefulness of the research. Some of these works, not cited and not critically analysed by the authors with respect to their work, are the following:

M. Roskosz, M. Bieniek / NDT&E International 54 (2013) 63–68

K. Yaoetal./JournalofMagnetismandMagneticMaterials354(2014)112–118

Z.D. Wang et al. / NDT&E International 43 (2010) 513–518

M. Roskosz, M. Bieniek / NDT&E International 45 (2012) 55–62

Zhang, Tan and Zheng, Sensors 2017, 17, 608

S. Bao, X. Liu and D. Zhang, Strain (2015) 51, 370–378

2) With reference to line 33, which are the numerous papers referred to for the selection of the criterion? The references are not cited and there is no critical analysis that justifies the choice.

3) In section 4, concerning the validation and numerical simulation, it would be useful to describe in detail the characteristics of the implemented FEM model. Without this information it is not possible to judge the reliability of the model and therefore the reliability of the validation.

4) The figures, in particular the numbers 6 to 9, should be equipped with a legend, a data label and above all an indication of the abscissa and ordinate quantities.

5) All acronyms present in the text should be reported in full the first time they are mentioned (eg RFM, RPM, SCZ, etc.)

Author Response

The entire text has been revised and the Reviewer’s suggestions have been taken into account. An explanation related to the experimental and numerical verification has been added.

Appropriate changes have been made following the comments and suggestions of the other reviewers.

The paper concerns many years of testing by means of the RMF technique of a large-size structure in the form of a gantry crane. As it turned out, the crane was in a complex strain-and-stress state. The values of the tangential component gradient measured in stress concentration zones exceeded 14A/m/mm and the gradient distribution was asymmetric.  The findings were confirmed by experimental and numerical verification.

1) The description of the state of the art is meagre, the bibliographical references are few and limited, especially those related to the theoretical aspects of RFM applied in the section 2. There is also a lack of references to important works in the same field, which would help to understand well the topic covered and the usefulness of the research. Some of these works, not cited and not critically analysed by the authors with respect to their work, are the following:

M. Roskosz, M. Bieniek / NDT&E International 54 (2013) 63–68

K. Yaoetal./JournalofMagnetismandMagneticMaterials354(2014)112–118

Z.D. Wang et al. / NDT&E International 43 (2010) 513–518

M. Roskosz, M. Bieniek / NDT&E International 45 (2012) 55–62

Zhang, Tan and Zheng, Sensors 2017, 17, 608

S. Bao, X. Liu and D. Zhang, Strain (2015) 51, 370–378

The description of the state of the art has been completed with proper reference to the issues analysed in the paper and following the suggestions of the other reviewers.

2) With reference to line 33, which are the numerous papers referred to for the selection of the criterion? The references are not cited and there is no critical analysis that justifies the choice.

The reference literature list has been updated. Instead of items 1 and 3, new and up-to-date references have been introduced. Item 2 concerns the testing of pipelines, i.e. large-size structures. 

Pipeline testing using the magnetic flux leakage (MFL) signals is presented in [ ]. Attention is drawn to the fact that the speed at which the inspection is carried out can reduce the signal value.

The criterion selection was determined by the fact that the testing concerned large-size structures and not laboratory samples.

3) In section 4, concerning the validation and numerical simulation, it would be useful to describe in detail the characteristics of the implemented FEM model. Without this information it is not possible to judge the reliability of the model and therefore the reliability of the validation.

 A detailed description of the FEM model has been added.

Round 2

Reviewer 3 Report

The paper has been improved and , to my opinion, is now suitable for publication.